# Liver transplantation in patients with a history of migration—A German single center comparative analysis

Julian Nikolaus Bucher[1]*, Maximilian Koenig[1], Markus Bo Schoenberg[1], Alexander Crispin[2], Michael Thomas[1], Martin Kurt Angele[1], Daniela Eser-Valeri[3], Alexander Lutz Gerbes[4,5,6], Jens Werner[1], Markus Otto Guba[1,5,6]

1 Department of General, Visceral and Transplant Surgery, Ludwig-Maximilians-University Munich, Munich, Germany, 2 Institute of Medical Informatics, Biometry and Epidemiology, Ludwig-Maximilians-University Munich, Munich, Germany, 3 Department of Psychiatry, Ludwig-Maximilians-University Munich, Munich, Germany, 4 Department of Medicine 2, Ludwig-Maximilians-University Munich, Munich, Germany, 5 Transplantation Centre Munich, Ludwig-Maximilians-University Munich, Munich, Germany, 6 Liver Centre Munich, Ludwig-Maximilians-University Munich, Munich, Germany

* julian.bucher@med.uni-muenchen.de

**Data Availability Statement:** Data cannot be shared publicly because although ananymized, our full data set might allow for the identification of individuals of the cohort from the available

## Abstract

Liver transplant (LT) programs in Germany increasingly face a multiethnic patient population. To date no outcome data for LT in patients with a history of migration is available for Germany. This complicates decision-making before wait-listing such patients. We conducted a single-center cohort analysis of all primary LT between April 2007 and December 2015, stratified for the history of migration to investigate differences in the outcome. We found transplant rates resembling the proportion of persons with a history of migration in the general public in the region of our center. Differences were found concerning age at LT and prevalence of underlying diseases. Re-Transplant rates, Kaplan-Meier Estimates for overall survival, also after stratification for viral hepatitis, sex, ethnicity or presence of a language-barrier showed no statistical differences. The multivariate analysis showed no migration-related covariate associated with a negative outcome. These results stand in contrast to most of the previous evidence from North America and the UK and need to be taken into consideration during the wait-listing process of patients with a history of migration in need of a LT in centers in the Eurotransplant region.

## Introduction

Persons with a history of migration are at higher risk of poverty and social exclusion, and there is evidence that they sometimes do not receive the care that best responds to their needs [1, 2]. As the population of central Europe increasingly develops in to a multi ethnic society with a strong influx of immigration, mostly from economically less developed countries, liver-transplant programs increasingly are confronted with immigrants in potential need of a liver transplant (LT) [3].

parameters. Data are available from the Ethics Committee (contact via ethikkommission@med. uni-muenchen.de) for researchers who meet the criteria for access to confidential data.

**Funding:** This work was funded by the Friedrich Baur research fund which was awarded to JNB (award-number 30/16). This intra-institutional funding for junior scientists was solely used to pay the institution's ethic committee processing fees. The funders had no role in study design, data collection and analysis, decision to publish, or preparation of the manuscript.

**Competing interests:** The authors have declared that no competing interests exist.

**Abbreviations:** HCC, Hepatocellular carcinoma; IQR, inter-quartile range; KMC, Kaplan-Meier curve; LT, liver transplantation; MELD, Model of Endstage Liver Disease (surrogate parameter for severity of liver disease); mig-group, group of patients with a history of migration; nonmig-group, group of patients without a history of migration; reLT, liver re-transplantation (liver transplantation after previous liver transplantation); RKI, Robert Koch Institute (institute for disease control and prevention of the Federal Republic of Germany); RR, Hazard rate ratio; UNOS, United Network for Organ Sharing (US organ allocation organization).

Besides the medical urgency, the chances for the long-term success of a LT have to be taken into consideration before placing the individual patient on the waiting list. The MELD score is an objective surrogate parameter for the medical urgency [4] whereas the assessment of the chances for long-term success has to be performed individually, based partly on evidence and mostly on the subjective expertise of the transplant specialists responsible for listing.

Evidence from North America shows that social, ethnic and economic factors can influence the outcome after solid organ transplantation [5–7]. In Europe, data on this subject is rather scarce, and the few studies with comparable objectives mostly examined kidney transplantation and came either from the UK, the Netherlands or Hungary [8–13]. Only one study from the UK from 1993 examined a history of migration as an outcome-relevant factor after liver transplantation and found inferior results in non-European immigrants [14].

As demographics and the ethnic composition of immigrant and minority populations in central Europe differ from the UK and North America and health care systems and social welfare programs vary significantly [4], conclusions from the presently available literature cannot be extrapolated for the German situation.

To provide evidence as support for the difficult process of evaluating the chances of long-term success of LT in patients with a history of migration we conducted a comparative analysis of liver transplant recipients with a history of migration at our center. We also searched for factors that could explain differences in outcomes and could be modified to improve care for this possibly disadvantaged population.

## Methods

We conducted a retrospective analysis of the prospective LT database of our center. We only included primary LT performed from April 2007 until December 2015. Patients who received a liver re-transplantation (reLT) for chronic allograft failure of an allograft transplanted before the studied period were excluded from the analysis. Also pediatric LT below the age of 14 years were excluded.

Included patients were assigned with a categorized migration status as defined by the Robert Koch Institute (RKI) [15], Germany's federal institute for disease-control and prevention: Patients with German citizenship who were born in Germany but have migrated parents were assigned to Group One. Patients with German citizenship who were born outside of Germany were assigned to Group Two. Patients who were born with German citizenship outside of Germany and migrated after birth (mostly resettlers of German descent from former enclaves in the former USSR) were assigned to Group Three. Patients with dual citizenship status, who were born in Germany with at least one parent who migrated to the country were assigned to Group Four. Patients with dual citizenship who were born abroad and immigrated after birth were assigned to Group Five. Patients with non-German citizenship and a registered residence in Germany were assigned to Group Six. Patients who immigrated to Germany with a limited residence authorization (including asylum-seekers and refugees) were assigned to Group Seven. Patients without a legal residence authorization were assigned to Group Eight. Patients who received the transplant during a tourist visit were assigned to Group Nine. Patients whose immigration status could not be identified were assigned to Group Ten.

### Parameters of interest

We analyzed overall survival as primary outcome parameter. Age at transplantation, indications and MELD scores at time of LT were included in the analysis. ReLT was categorized as re-LT for primary allograft failure if it was performed within the same hospital stay, and as chronic allograft failure if reLT was performed after readmission.

To investigate migration-related factors in pre- and post-transplant patient management, we looked at patient mobility, proficiency in the German language according to the ILR Scale [16] and perceived quality of communication with the medical staff at our transplant clinic. If patients could not be contacted, information was obtained from first degree family-members or primary physicians. All patients gave their informed consent for the use of anonymized medical data for analysis and publication. The study was approved by the institution's ethics committee (Ethikkommission bei der LMU, Ref.Nr: 519–16).

Continuous data are presented as median and IQR, and differences between two groups were determined using the Mann-Whitney U test. Categorical data are presented as frequency of occurrence and the two-tailed Fisher's exact test was used to compare different groups. Patient survival was determined by Kaplan-Meier estimators (Kaplan-Meier curves, KMC) and cumulative incidences of censored events were compared by log-rank tests. For the in depth analysis of the mig-group concerning German citizenship, 8 patients with a history of migration had to be excluded from the analysis for unknown citizenship status at time of transplant. Cox proportional hazards regression analysis was used to adjust survival for potential confounding by known risk factors for mortality. Predictor variables (migration background, German citizenship at transplantation, age, sex, indication for LT, labMELD at transplantation and type of organ allocation (MELD-based- vs. rescue- vs. high-urgency-allocation)) were included in the model using forward-selection based on p-values from likelihood-ratio tests. For all analyses we considered p-values $\leq 0.05$ to be statistically significant. Due to the exploratory nature of our analysis we did not adjust the alpha-level for multiple testing. Statistical analysis was performed with SPSS for Windows release 24 (IBM, Armonk, USA).

## Results

### Descriptive analysis

From April 2007 until December 2015 a total of 417 LT were performed at our center. 358 (86%) were primary LT, 28 (7%) were early reLT and 31 (7%) were reLT for chronic allograft failure. Of these reLT for chronic allograft failure, 12 patients had received the first transplant before the studied period and were excluded from further analysis.

67 out of 358 patients who received a primary LT (19%) had a history of migration (further referred to as 'mig-group'; categorized migration status: Group 1: n = 8; Group 2: n = 6; Group 3: n = 6; Group 4: n = 1; Group 5: n = 4; Group 6: n = 30; Group 7: n = 2; Group 8: n = 0: Group 9: n = 2; Group 10: n = 8). In the mig-group 11 patients received a reLT while 31 patients without a history of migration (further referred to as 'nonmig-group') were retransplanted (16% vs. 11%; p = 0.206). In the mig-group 8 patients underwent reLT for primary allograft failure vs. 19 patients in the nonmig-group (12% vs. 7%; p = 0.131). ReLT for chronic allograft failure had to be performed in 3 patients of the mig-group and in 12 patients of the nonmig-group (4% vs. 4%; p = 1.000). Median age at LT was 53.3 years (IQR 12.76) with the median age in patients with migration background being significantly lower than in patients without migration background (49.1 (IQR 19.34) vs. 54.2 (IQR 11.44) years; p = 0.001). This age discrepancy was most evident in the subgroup of female patients with migration background who were transplanted at a median age of 41.2 (IQR 15.9) years compared to the median age of females in the nonmig-group of 52.4 (IQR 12.91) years (p = 0.007). In both groups the ratio of female vs. male patients was approximately 1:2 (p = 0.7747). No differences were noted in medians of allocation- and lab-MELD scores at time of LT, standard and non-standard exception MELD scores or relative numbers of granted standard and non-standard exception status in both groups. Also the prevalence of standard- and high-urgency allocations and rescue-allocations was similar (*see* Table 1).

**Table 1. Characteristics of all included liver transplant recipients (overall), recipients without- (nonmig-group), and with migration background (mig-group).**

| Recipient characteristics | overall (n = 358) | no history of migration (n = 291) | with history of migration (n = 67) | p |
|---|---|---|---|---|
| median age (IQR) | 53.3 (12.76) | 54.2 (11.44) | 49.1 (19.34) | **p = 0.0011** |
| median age female patients (IQR) | 50.7 (15.87) | 52.4 (12.91) | 41.2 (15.9) | **p = 0.0069** |
| median age male patients (IQR) | 54.3 (11.42) | 54.9 (10.25) | 50.5 (18.45) | **p = 0.0334** |
| sex (F/M) | 119/239 | 98/193 | 21/46 | p = 0.7747 |
| MELD-score at time of transplant | | | | |
| median allocation MELD-score (IQR) | 28 (12.25) | 28 (13) | 27 (10) | p = 0.3954 |
| median labMELD-score (IQR) | 20 (21) | 21 (20) | 18 (17) | p = 0.2162 |
| (N)SE MELD (IQR) | 26 (5) | 25 (4) | 27 (7) | p = 0.5895 |
| standard exception (%) | 119 (33%) | 93 (32%) | 26 (39%) | p = 0.4576 |
| non standard exception [NSE] (%) | 11 (3%) | 10 (3%) | 1 (2%) | |
| type of allocation allocation | | | | |
| No. of standard allocations (%) | 186 (52%) | 153 (53%) | 33 (49%) | p = 0.240 |
| No. of high urgency status allocations (%) | 36 (10%) | 29 (10%) | 7 (11%) | |
| No. of rescue-allocations (%) | 136 (38%) | 109 (38%) | 27 (40%) | |
| indications for liver transplantation | | | | |
| acute liver failure (%) | 26 (7%) | 19 (7%) | 7 (10%) | p = 0.2948 |
| alcoholic cirrhosis (%) | 66 (18%) | 59 (20%) | 7 (10%) | p = 0.0793 |
| NASH / NAFLD (%) | 1 (<1%) | 1 (<1%) | 0 (0%) | p = 1 |
| cryptogenic Cirrhosis (%) | 24 (7%) | 19 (7%) | 5 (7%) | p = 0.7872 |
| viral hepatitis (%) | 52 (15%) | 38 (13%) | 14 (21%) | p = 0.1227 |
| HepB hepatitis (%) | 8 (2%) | 7 (2%) | 1 (1%) | p = 1 |
| HepBD hepatitis (%) | 15 (4%) | 6 (2%) | 9 (13%) | **p = 0.0003** |
| HepC hepatitis (%) | 29 (8%) | 25 (9%) | 4 (6%) | p = 0.6227 |
| viral hepatitis [as underlying disease (including HCC)] (%) | 110 (31%) | 78 (27%) | 32 (48%) | **p = 0.0012** |
| Hep B hepatitis (%) [as underlying disease (including HCC)] (%) | 19 (5%) | 11 (4%) | 8 (12%) | **p = 0.0134** |
| Hep BD hepatitis [as underlying disease (including HCC)] (%) | 18 (5%) | 6 (2%) | 12 (18%) | **p = 0.0001** |
| Hep C hepatitis [as underlying disease (including HCC)] (%) | 73 (20%) | 61 (21%) | 12 (18%) | p = 0.6189 |
| alcoholic cirrhosis [as underlying disease (including HCC)] (%) | 96 (27%) | 87 (30%) | 9 (13%) | **p = 0.0089** |
| cryptogenic cirrhosis [as underlying disease (including HCC)] (%) | 31 (9%) | 24 (8%) | 7 (10%) | p = 0.6291 |
| HCC (%) | 101 (28%) | 79 (27%) | 22 (33%) | p = 0.3681 |
| HCC in alcoholic cirrhosis (%) | 30 (8%) | 28 (10%) | 2 (3%) | p = 0.0886 |
| HCC in viral hepatitis (%) | 58 (16%) | 40 (14%) | 18 (27%) | **p = 0.011** |
| autoimmune Hepatitis (%) | 11 (3%) | 10 (3%) | 1 (1%) | p = 0.4882 |
| Cholestatic liver disease | 41 (11%) | 35 (12%) | 5 (7%) | p = 0.3899 |
| PSC (%) | 24 (7%) | 20 (7%) | 4 (6%) | p = 1 |
| PBC (%) | 6 (2%) | 5 (2%) | 1 (1%) | p = 1 |
| SSC (%) | 11 (3%) | 10 (3%) | 1 (1%) | p = 0.4882 |
| metabolic/genetic disorders | 14 (4%) | 12 (4%) | 2 (3%) | p = 1 |
| M. Wilson (%) | 2 (1%) | 1 (<1%) | 1 (1%) | p = 1 |
| Hemocromatosis (%) | 2 (1%) | 2 (1%) | 0 (0%) | p = 1 |
| other metabolic/genetic disorders (%) | 10 (3%) | 9 (3%) | 1 (1%) | p = 0.6952 |
| Cystic liver diesease (%) | 5 (1%) | 4 (1%) | 1 (1%) | p = 1 |
| Echinococcosis (%) | 1 (<1%) | 1 (<1%) | 0 (0%) | p = 1 |
| other liver tumors (%) | 6 (2%) | 5 (2%) | 1 (1%) | p = 1 |
| CCC (%) | 2 (1%) | 2 (1%) | 0 (0%) | p = 1 |

*(Continued)*

**Table 1.** (Continued)

| Recipient characteristics | overall (n = 358) | no history of migration (n = 291) | with history of migration (n = 67) | *p* |
|---|---|---|---|---|
| Budd-Chiari (%) | 5 (1%) | 4 (1%) | 1 (1%) | p = 1 |
| other liver disease (%) | 3 (%) | 3 (1%) | 0 (0%) | p = 1 |

Data are median (IQR) or n (%).

The descriptive analysis of the mig-group showed that 25 patients were German citizens at time of primary LT (37%; migration status groups one to five; including dual citizenship), 34 patients were non-German citizens (51%; groups six to nine) and in eight patients the migration-status could not be categorized (12%; group ten). Of 59 patients with known migration status, 20 were born in Western Europe (Germany, Greece, Italy and Spain; 34%), 24 were born in Eastern Europe (Poland, Hungary, Croatia, Romania, Czech Republic, Kosovo and Turkey; 41%) and 15 were born in countries outside of Continental Europe (25%). In 59 patients of the mig-group we could evaluate the proficiency of German language: 42 patients (71%) were excellent, very good or good speakers while 17 patients (29%) spoke fair or poor. In 57 of these 59 patients we were able to obtain information about the subjective quality of communication with the medical staff in our transplant clinic: three patients had difficulties in communication because of a language barrier (5%), of which two patients had difficulties in understanding spoken, therapy relevant information (3%). All other patients stated not to have difficulties in communication with our medical staff.

Indications for LT and underlying disease differed significantly between the two groups: Alcoholic cirrhosis was more prevalent in the nonmig-group with 29.9% compared to 13.4% in the mig-group (p = 0.009). Prevalence of viral hepatitis as underlying disease was higher in the mig-group with 47.8% compared to 26.8% in the nonmig-group (p = 0.001) with the highest difference evident in hepatitis BD co-infection with 17.9% vs. 2.1% in the mig- vs. the nonmig-group (p < 0.0001). The prevalence of HCC in cirrhosis and non-cirrhosis was similar in both groups while viral hepatitis as underlying disease for HCC was more prevalent in the mig-group (50.6% vs. 81.8% nonmig- vs. mig-group; p = 0.012).

## Survival analysis by migration status

The one year survival-rate was 74% in the nonmig group and 86% in the mig-group (p = 0.1621). The KMC analysis of overall survival showed no difference in 5-year survival between the mig-group and the nonmig-group (p = 0.54) (*see* Fig 1). In the subgroup-analyses for patients with viral-hepatitis as underlying disease, and for patients suffering from HCC we also saw no differences in long-term survival between the groups (p = 0.93 and p = 0.577 respectively) (*see* Fig 2). When we stratified the mig- and nonmig-group for sex, we saw a better long-term survival curve in female patients of the mig-group compared to males of the mig-group and also compared to the nonmig-group, yet without statistical significance (p = 0.49) (*see* Fig 3). The in-depth descriptive analysis of female patients of the mig- and the nonmig-group showed a younger median age in females of the mig group (41.2 (IQR 15.9) vs. 52.4 (IQR 12.91) years; p = 0.007) and a tendency to lower allocation and labMELD scores at time of LT (median allocation MELD 29 (IQR 12.25) vs. 26.5 (IQR 18.0) (p = 0.091) and median labMELD 27 (IQR 21.0) vs. 17 (IQR 17.5), p = 0.095; female nonmig-group vs. female mig-group respectively). Other outcome relevant parameters at LT were not different (see Table 1).

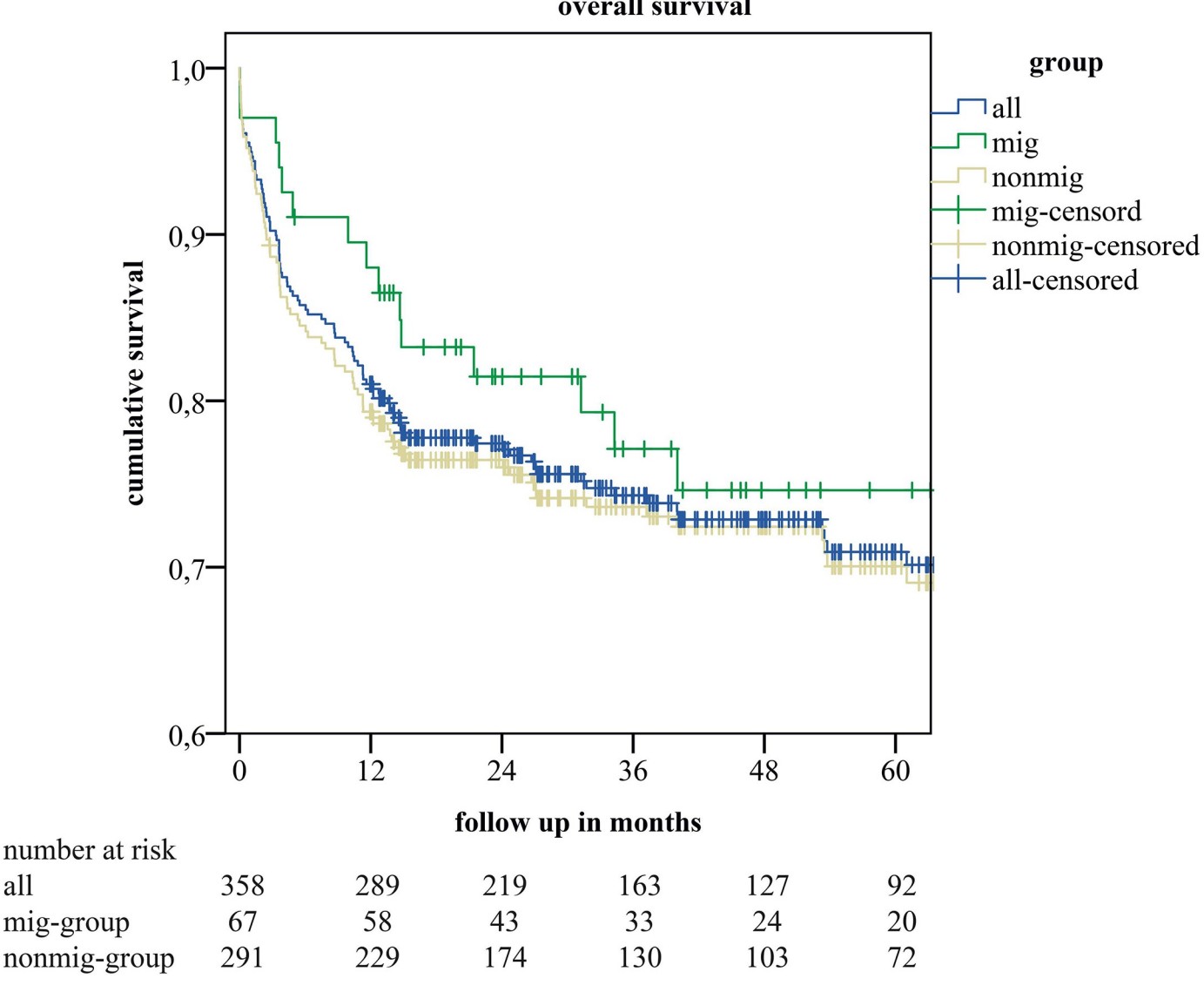

**Fig 1. Kaplan-Meier estimates of cumulative survival of all included recipients of a liver transplant (all), recipients without- (nonmig-group), and with migration background (mig-group).**

The Cox regression analysis with the mig-group stratified for migration-status according to the RKI-definition into patients that were German citizens (Migration-status Groups One-Five) at time of LT (mig 1) or who were not German citizens (Migration-status Groups Six-Nine) at time of LT (mig2) revealed an association of the labMELD score at LT and overall survival (RR = 1.025, 95% CI 1.006–1.044, p = 0.009). Also we discovered an association of German citizenship at time of transplant in patients with a history of migration with overall survival (RR = 0.117, 95% CI 0.016–0.841, p = 0.033). Patients of the mig 1 group were significantly younger than non-mig patients at transplantation (43.73 years (IQR 22.57) vs. 54.24 (IQR 11.31) p = 0.005), while they were not significantly younger than mig patients without citizenship at time of transplant (49.48 years (IQR 15.82) p = 0.226). Mig 2 patients were significantly younger than non-mig patients (p = 0.034). When we performed a KMC analysis of the mig-group stratified for migration-status (mig 1 vs. mig 2 vs. non mig)we saw a better 5-year

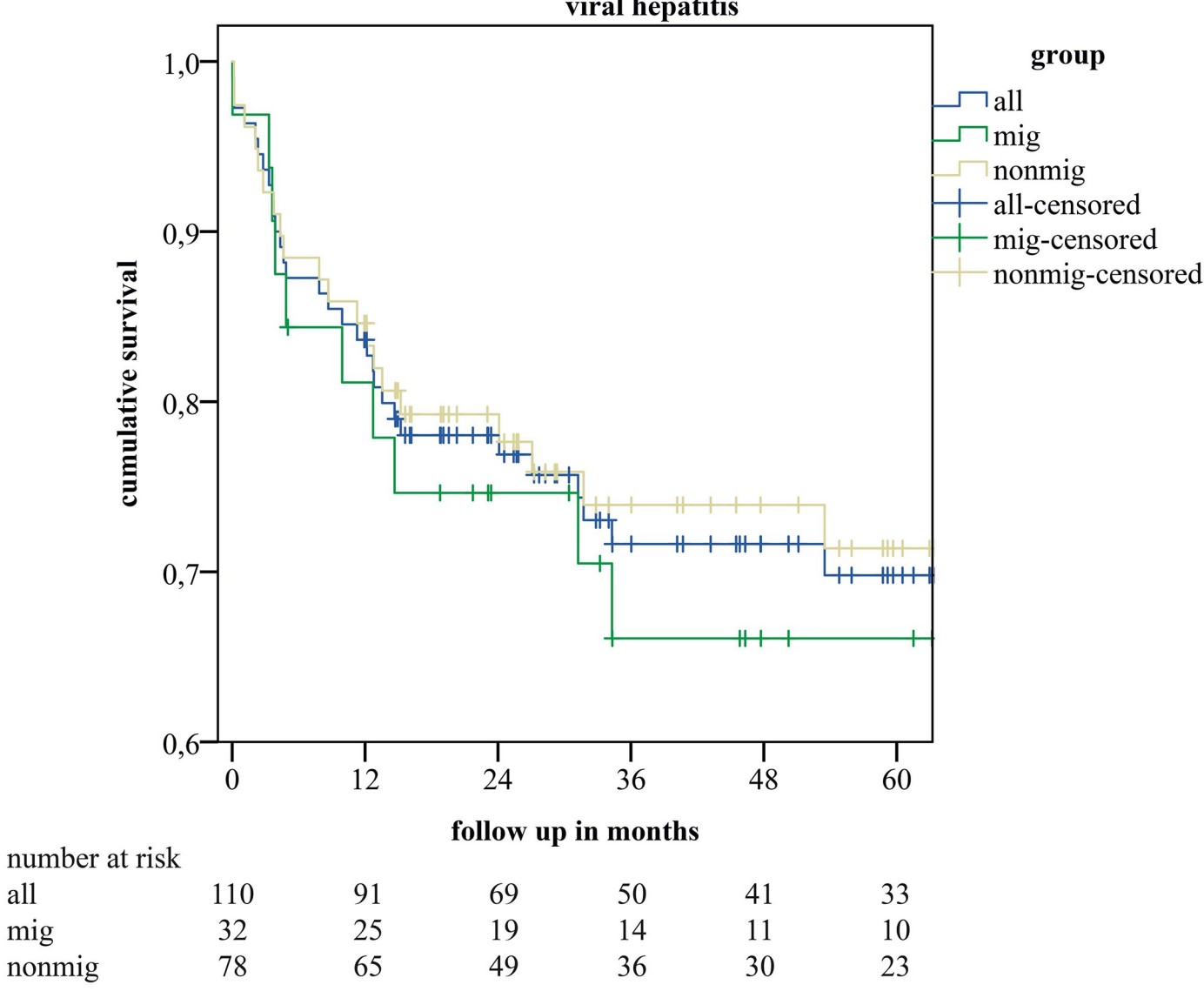

**Fig 2. Kaplan-Meier estimates of cumulative survival for the subgroups with viral hepatitis as underlying disease of all included recipients of a liver transplant (all), recipients without- (nonmig-group), and with migration background (mig-group).**

survival in the mig1 subgroup (p = 0.04). However, the survival curve of patients of the mig2 subgroup was similar to the survival curve or the nonmig-group without any statistical difference (*see* Fig 3). We also assessed the effect of language-barrier on survival after LT within the mig-group by comparing KMCs of the patients with excellent, very-good and good proficiency in German (lb1-group) with basic or poor speakers (lb2-group) without any differences in 5-year survival (p = 0.213) (*see* Fig 4).

## Discussion

Studies that examined the outcome after solid organ transplantation in European and non-European immigrants have previously been conducted in the UK, in the Netherlands and Hungary [8–13]. Most of these studies investigated kidney transplantation and found an

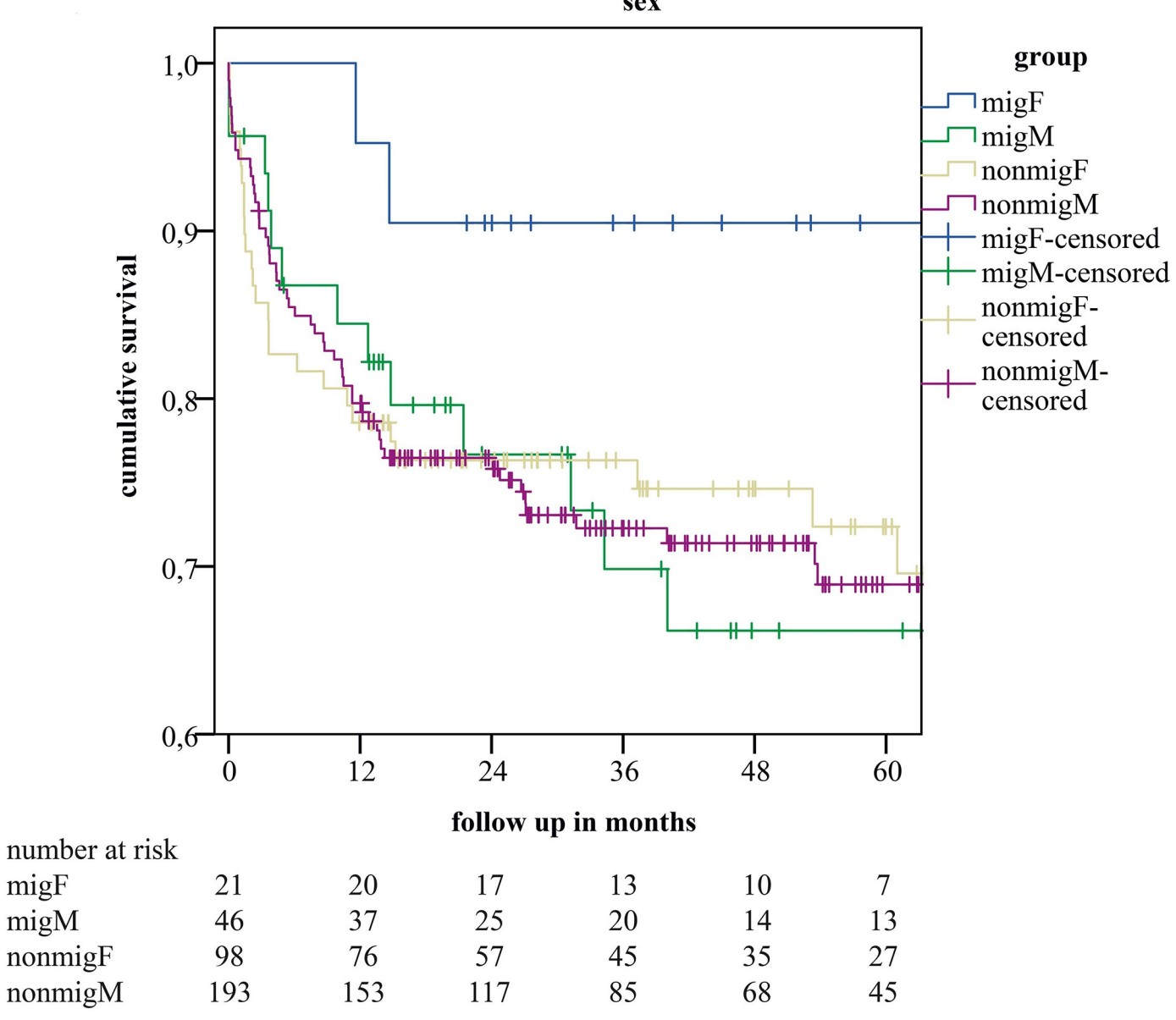

**Fig 3. Kaplan-Meier estimates of cumulative survival after stratification for sex in female- (migF) and male recipients (migM) with migration background, and female- (nonmigF) and male recipients (nonmigM) without migration background.**

inferior graft- and patient survival in recipients with a history of migration. Just one study from the UK in 1993 compared the outcome of LT in Europeans and non-European immigrants and found an inferior short and long-term survival in migrated patients [14]. To this date, no study with a comparable objective has been conducted in central Europe. Facing the current increase of immigration, and the prevailing situation of an increasing number of immigrants in need of solid organ transplantation [3] we decided to update the data on this matter from a German perspective.

Our Transplant Center is located in the south-east of Germany in an economically prosperous region with an annual gross domestic product of around 81.000 € per capita [17]. A microcensus in 2008 showed that 15.8–23.5% of the population had a history of migration, a

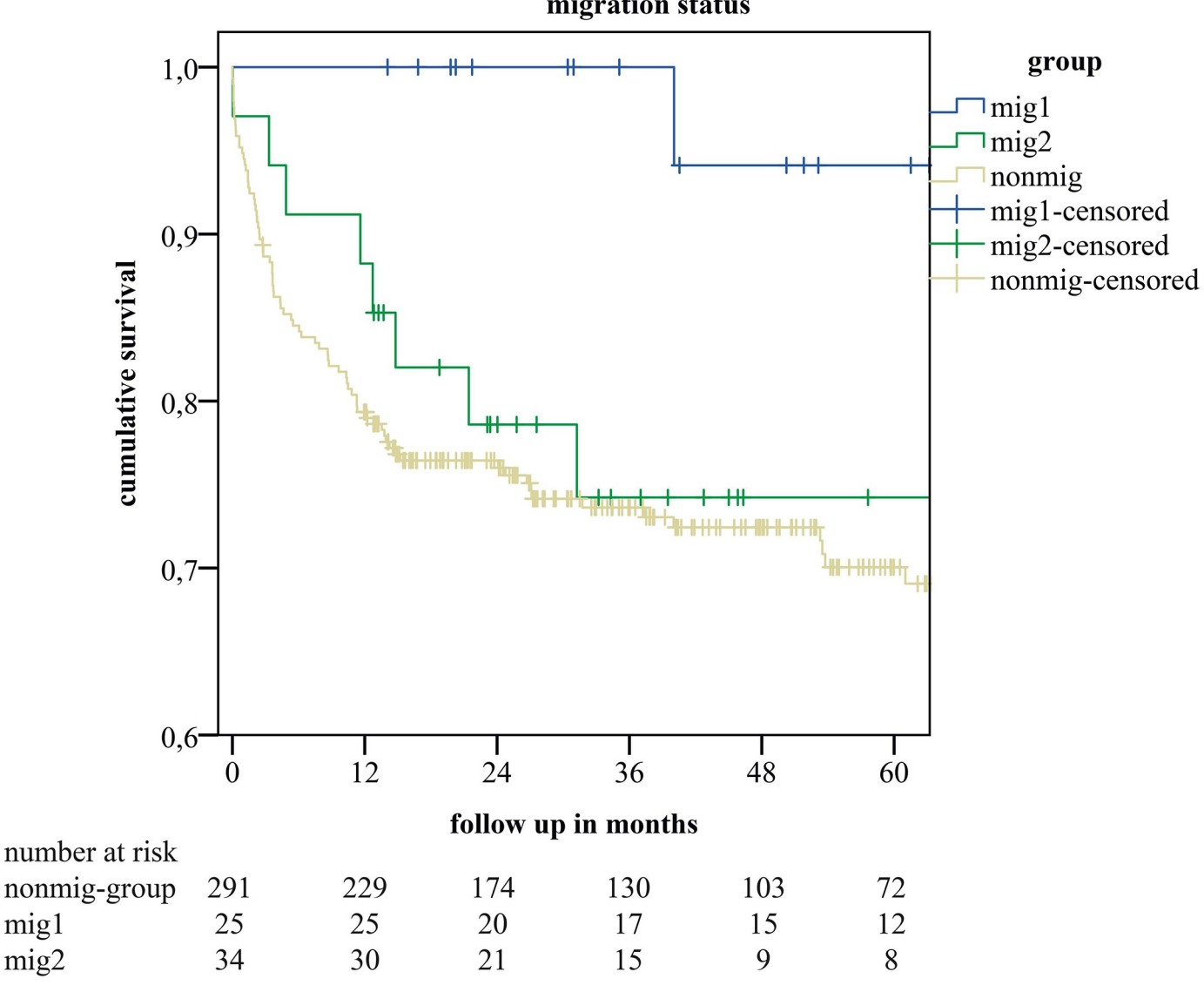

**Fig 4. Kaplan-Meier estimates of cumulative survival in recipients with migration background stratified for migration status (mig1—German citizenship at time of transplant vs. mig2—no German citizenship at time of transplant) and recipients without migration background (nonmig).**

proportion slightly higher than the national average of 16.1% [18]. One third of the persons with a history of migration in our region were non-EU immigrants, while 20% were immigrants from EU-member countries. Further 20% were persons who were born in Germany but had at least one parent who had migrated to Germany. The remaining third were resettlers of German descent or persons who became naturalized after the Second World War [18]. We found a similar composition in the cohort with a history of migration at our center (see Fig 5). A differentiated view on the population with a history of migration concerning age-distribution showed that the proportion of immigrants in younger age groups was substantially higher than in older age-groups, producing a pyramid shaped population pyramid [18], while the overall-German population pyramid is mushroom-shaped.

The descriptive analysis of 358 patients who received a primary LT at our center in the studied period showed that 67 (18.7%) had a history of migration. This proportion is equivalent to

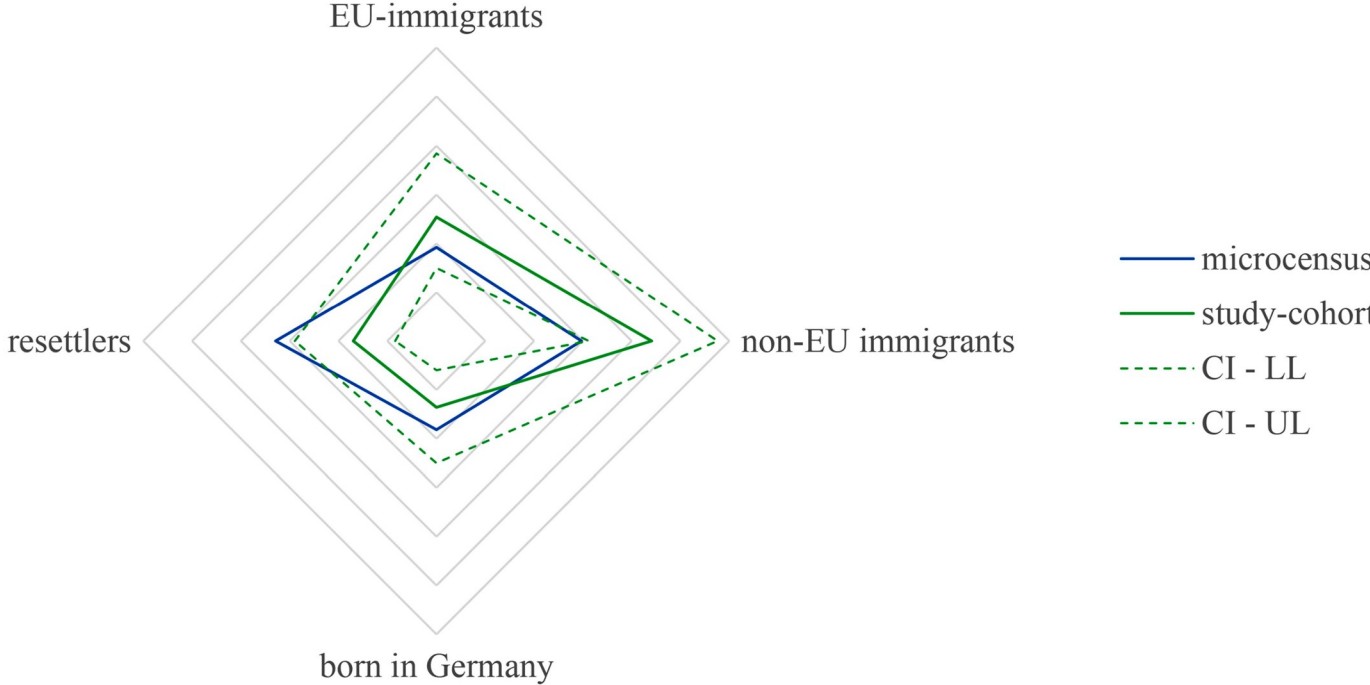

**Fig 5. Compositions of the population with migration background in the region of our transplant center according to the microcensus 2008 and the studied cohort of liver transplant recipients with migration background at our center (dotted lines indicate the upper (UL) and lower limit (LL) of the 95% Confidence Interval).** In the cohort of LT recipients at our transplant center, 13.6% were born in Germany (95% CI 6–25%), 17% were resettlers or became naturalized after the Second World War (95% CI 8.4–29%), 25.4% were immigrants from EU-member countries (95% CI 15–38.4) and 44.1% were immigrants from non-EU countries (95% CI 31.2–57.6%).

the population in the region served by our transplant center where 18.6% of the people have a history of migration [19].

The prevalence of disease leading to the need for LT differed significantly between the groups: While alcoholic cirrhosis was the most prevalent disease in patients without migration background (29.9%) it was viral hepatitis in patients with a history of migration (47.8%). These proportions are most likely attributable to differences in lifestyle, cultural background, and especially to the disease prevalence in the country of origin and to access to preventive measures like e.g. vaccination [20] between the migrant- and non-migrant population in Germany [21–24].

We found a significantly younger median age at transplantation in patients with a history of migration compared with non-migrated patients (49.1 vs. 54.2 years; p ≤ 0.001174). Besides the pyramid-shaped pattern of age distribution in the migrant population in our region, with the majority of persons being younger than 50 years [18] and the predominance of vertical transmission of viral hepatitis [25], this could also be explained by a generally impaired access to healthcare and underutilization of specialist-care in the migrant population leading to an increased disease severity at younger age and at diagnosis [26, 27]. One could further speculate that the significantly younger age at transplantation in the subgroup of female patients with a history of migration is a result of a socio-cultural disadvantage of women in some ethnic groups leading to even later first referral in the course of disease.

The survival analysis showed no significant differences in short- and long-term overall survival between the groups. We even saw a tendency to better long-term survival in the subgroup of female recipients with a history of migration which is most likely attributable to the

significantly younger age at transplantation in this group. To exclude confounders from the different epidemiology, we performed a subgroup analysis of patients with viral hepatitis as underlying disease for LT. This analysis did also not show a difference in short- and longterm survival. The multivariate Cox-regression analysis of covariates that could be associated with an inferior outcome did not show an association of migration background with negative outcomes. To further investigate a history of migration as a risk factor for the outcome after LT we performed a subgroup analysis of the mig-group, stratified for migration status. We found that patients with a history of migration who were German citizens at time of transplant had a significantly better survival than patients without a history of migration or without German citizenship at time of transplant. The survival of patients with a history of migration who were not German citizens at time of transplant was however not inferior compared to the survival of patients without a history of migration. The protective effect of German citizenship at time of transplant in patients with a history of migration might be explainable by the significantly younger age at transplantation in this subgroup. It also might resemble a state of better and longer-lasting integration, that could have effects through the better orientation and functionality of these patients in the German healthcare system. These explanations however are not sufficiently backed by the available data and are just speculative.

While a history of migration is an unalterable factor, language barrier is an alterable factor and less-effective communication with migrant patients can cause misunderstandings and non-adherence to treatment [28, 29]. We investigated whether patients with an existing language barrier had an impaired survival after LT. We found no association of language barrier with impaired survival. The evaluation of the perceived quality of communication with the medical professionals at our transplant clinic showed that the vast majority of patients with a history of migration perceived the quality of communication to be excellent or good and only 4% reported to have difficulties in understanding therapy-relevant information. We interpret these data as an indicator for a sufficiently individualized care provided to patients with a history of migration despite an existing language barrier. Access to high-quality post-transplant care is also dependent on an unimpaired access to first hand care at the transplant clinic. In hematopoetic stem cell transplantation, geographical distance to the transplant center is associated with an impaired outcome after transplantation [30]. Similar results have been shown for liver transplant candidates who have impaired access to the waitlist, lower chance of being transplanted and higher mortality if living more than 100 miles from the transplant center [31]. Only one patient of the mig-group reported to have difficulties with transportation from home to our transplant center and we found no association of distance to the transplant center with mortality in this cohort.

Outcomes after LT can be influenced by insurance status. DuBay et al. found an inferior outcome in Medicare and Medicaid dependent LT-recipients compared with privately insured recipients in a large Scientific Registry of Transplant Recipients analysis [32]. As health insurance is compulsory in Germany and every registered resident is granted the coverage of state-of-the art medical treatment of life-threatening diseases [33], all patients in the studied cohort had sufficient cost-coverage at time of LT. In the case of liver transplantation every type of German health-insurance also covers costs for stationary and ambulatory medical rehabilitation and transport costs for on-site medical follow-up at the transplant center. In the rare case of unregistered patients presenting in the acute need of a liver transplant, an emergency protocol is sent to the social services department and the patient will receive the care he/she needs. Financial compensation is then sorted out during or after the transplant and as a last resort option, the social service department will compensate for the treatment. However, we haven't experienced this situation at our unit yet and only are aware of some cases of acute liver failure in unregistered asylum seekers where the course of action mentioned above had to be taken

[3]. We therefore conclude that the universal healthcare and social welfare-system in Germany mitigates the problems of migrants undergoing complex medical treatments such as LT.

## Conclusion

We found no inferior outcome in liver transplant recipients with a history of migration compared with indigenous recipients at our center. Although the interpretation of our data is limited by its single-centered nature, these results stand in contrast to most of the previous evidence. The presumption that a migration background is a risk factor for the outcome after liver transplantation is not necessarily true for the German situation. This needs to be taken into consideration during the evaluation for waitlisting patients with migration background for liver transplantation in Germany. Prima vista the similar transplant rates and comparable outcomes also imply the absence of inequalities in access to LT and to high-quality post-transplant care. However, the prevalence and severity of liver disease grows incremental to social status. Further, social-, ethnic- and economic factors can influence the access to the transplant-waitlist as well as affect waitlist mortality [34]. Therefor these clues have to be drawn with caution and an intention-to-treat analysis at time of presentation to the transplant clinic, before evaluation for LT is needed.

## Acknowledgments

This work was funded by the Friedrich Baur research fund, an intra-institutional funding for junior scientists. We thank Prof. Alexandr Bazhin for his infrastructural support. We thank Mrs. Vivien Thiemann for supporting the data collection and the maintenance of the liver transplant data base. We thank the team of the Munich Transplant Center (TxM) for providing us with advice and anecdotal reports which were essential for the design and interpretation of this study.

## Author Contributions

**Conceptualization:** Julian Nikolaus Bucher, Alexander Crispin, Markus Otto Guba.

**Data curation:** Maximilian Koenig.

**Formal analysis:** Julian Nikolaus Bucher, Maximilian Koenig, Markus Bo Schoenberg, Alexander Crispin.

**Funding acquisition:** Jens Werner.

**Investigation:** Julian Nikolaus Bucher, Maximilian Koenig, Daniela Eser-Valeri, Alexander Lutz Gerbes, Markus Otto Guba.

**Methodology:** Julian Nikolaus Bucher, Markus Bo Schoenberg, Markus Otto Guba.

**Project administration:** Julian Nikolaus Bucher, Markus Otto Guba.

**Resources:** Jens Werner, Markus Otto Guba.

**Supervision:** Michael Thomas, Martin Kurt Angele, Alexander Lutz Gerbes, Jens Werner, Markus Otto Guba.

**Validation:** Julian Nikolaus Bucher, Markus Bo Schoenberg, Alexander Crispin, Michael Thomas, Daniela Eser-Valeri, Markus Otto Guba.

**Visualization:** Markus Bo Schoenberg.

**Writing – original draft:** Julian Nikolaus Bucher.

**Writing – review & editing:** Julian Nikolaus Bucher, Markus Bo Schoenberg, Alexander Crispin, Michael Thomas, Alexander Lutz Gerbes, Jens Werner, Markus Otto Guba.

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
