## [Decision Letter · Decision Letter 0]

27 Jul 2019

PONE-D-19-18755

Liver transplantation in patients with a history of migration – a German single center comparative analysis

PLOS ONE

Dear Bucher,

Thank you for submitting your manuscript to PLOS ONE. After careful consideration, we feel that it has merit but does not fully meet PLOS ONE’s publication criteria as it currently stands. Therefore, we invite you to submit a revised version of the manuscript that addresses the points raised during the review process.

ACADEMIC EDITOR: 

Intesting manuscript which was appreciated by both expert reviewers. They also came with constructive comments to improve the paper, and points that need to be clarified before we can consider publication.

Please address all issues in a point-by-point response.

We would appreciate receiving your revised manuscript by Sep 10 2019 11:59PM. To enhance the reproducibility of your results, we recommend that if applicable you deposit your laboratory protocols in protocols.io, where a protocol can be assigned its own identifier (DOI) such that it can be cited independently in the future. For instructions see: http://journals.plos.org/plosone/s/submission-guidelines#loc-laboratory-protocols

We look forward to receiving your revised manuscript.

Kind regards,

Frank JMF Dor, M.D., Ph.D.

Academic Editor

PLOS ONE

Journal Requirements:

1. Please amend your current ethics statement to address the following concerns: Please explain why was written consent was not obtained, how you recorded/documented participant consent, and if the ethics committees/IRBs approved this consent procedure.

Reviewers' comments:

Reviewer's Responses to Questions

**Comments to the Author**

1. Is the manuscript technically sound, and do the data support the conclusions?

Reviewer #1: Yes

Reviewer #2: Yes

2. Has the statistical analysis been performed appropriately and rigorously? 

Reviewer #1: Yes

Reviewer #2: Yes

3. Have the authors made all data underlying the findings in their manuscript fully available?

Reviewer #1: Yes

Reviewer #2: Yes

4. Is the manuscript presented in an intelligible fashion and written in standard English?

Reviewer #1: Yes

Reviewer #2: Yes

5. Review Comments to the Author

Reviewer #1: This is an extremely interesting paper looking at the results of Liver Tx in patients with a history of migration in a German center. This has been found a negative predictor of survival in US and UK cohorts but has not been studied in detail in a continental EU country. Shortlty speaking no difference was found in outcome. The most relevant difference was seen in indication and demographics: more alcoholic cirrhosis in the non migrant population and more viral hepatitis in the migrant group and a younger age at transplant in the migrant group. Authors are to be congratulated to discuss this sometimes controversial subject and for their work and results.

My major comment /question is that the equivalent results in these 2 populations might reflect an adequate patient selection independent of the migratory status. As a corollary question, I was wondering whether the percentage of LTx candidates turned down from transplant (not listed) was the same according to the migratory background? For example the percentage of patients with poor communication capacities was very small in the mig group. Poor communication is a general contraindication for LTx whatever the ethnical background. Was there a higher rate of patients excluded from LTx for poor communication and poor language skills in mig patients?

A note on the juridic aspect of LTx in migrants could be added. I suppose that by law, all registred migrants have access to Transplant and health care in general. The true difficulty is in patients who very recently arrived and are not resident yet or not registred yet. Were the authors confronted with this situation? Could they highlight this problematic? I remember that a few cases of LTx for ALF (I think for mushroom intoxication) in Germany were reported at a transplant meeting in Belgium (BTS) a few years ago with good outcome. Are the authors aware of these cases? Have they been published/reported/publicized (in literature and/or media)? If yes could it be possible to refer to them?

Another difficulty is the access to medications and follow up in case of planned return to a home land without German/European health care. Have the authors been confronted with this scenario and would this play a role in decision-making?

One might argue that german mig born in Germany might represent a lower risk population compared to recently arrived non german citizens? Can authors comment on that?

Within the mig group, german citizens mig did better than non german. However non german mig did equally well compared to non mig patients. What about german citizens mig versus non mig control?

The MV analysis showed no impact of mig status. Did the authors stratify this analysis depending upon the various mig types? German citizens vs non citizens mig etc?

Again congratulations to this team for this study and the results

Best Regards

Jacques Pirenne

Reviewer #2: In this manuscript Bucher et al. investigates the role of recipient history migration on the outcome of liver transplantation in a single center, descriptive cohort study. They did not find differences in outcome between recipient with and without the history of migration.

I read with interest this manuscript and I have few comments and questions to the authors:

The authors perform extensive statistical analysis on this rather small group of patients as 67 patients out of 358 patients had a history of migration. The authors divided them into 10 groups according to the RKI, however they did not give any number of patients assigned to each group. Could you provide this data? Later in their KM analysis they stratified these patients later into two groups.

7 patients from mig-group received liver transplantation for acute liver failure. Were in this group also patients who stayed in Germany temporary and moved to their countries after transplantation?

Did you analyse also graft survival in both groups? If so did you find any differences? It would be also good to know if the compliance after liver transplantation did not differ between two groups.

From your KM analysis (patients at risks) it seems that majority of mig-group has been transplanted in the later period of your study period. Do you have any explanation for this?

I would be very careful in extrapolating the result of this study on the German situation as the authors did in their conclusions. The retrospective and single-center nature of this study is a serious limitation, so it would be very interesting a have a national data on this topic, before drawing a conclusion that migration background is not a risk factor for the outcome in liver transplantation.

General comment:

As you perform extensive analysis of your relatively small data, the manuscript is not easy to read and its message is somehow lost in the text. I would shorten this manuscript and focus on the most important issues.

Additional comment:

In my opinion inclusion of Turkey in the Eastern Europe is not correct.

6. PLOS authors have the option to publish the peer review history of their article (what does this mean?). If published, this will include your full peer review and any attached files.

Reviewer #1: No

Reviewer #2: No

---

## [Author Response · Author response to Decision Letter 0]

22 Sep 2019

Answers to Reviewers questions point by point:

Reviewer #1: This is an extremely interesting paper looking at the results of Liver Tx in patients with a history of migration in a German center. This has been found a negative predictor of survival in US and UK cohorts but has not been studied in detail in a continental EU country. Shortlty speaking no difference was found in outcome. The most relevant difference was seen in indication and demographics: more alcoholic cirrhosis in the non migrant population and more viral hepatitis in the migrant group and a younger age at transplant in the migrant group. Authors are to be congratulated to discuss this sometimes controversial subject and for their work and results.

My major comment /question is that the equivalent results in these 2 populations might reflect an adequate patient selection independent of the migratory status. As a corollary question, I was wondering whether the percentage of LTx candidates turned down from transplant (not listed) was the same according to the migratory background? For example the percentage of patients with poor communication capacities was very small in the mig group. Poor communication is a general contraindication for LTx whatever the ethnical background. Was there a higher rate of patients excluded from LTx for poor communication and poor language skills in mig patients?

Answer: We share your view on the topic of patient selection as a confounder of our analysis. We therefore reviewed the available protocols of our interdisciplinary transplant board during the studied period. Only beginning in 2014 the reasons for the decline from the waiting list was documented centrally at our institution and available to us. Of 48 declines from wait-listing through our board, 16 patients had a history of migration (33%). Reasons for the decline from listing were strictly medical (2 for continued alcohol abuse, 4 for tumors not eligible for transplant, 6 without need for transplant, 2 deaths before listing and 2 too sick from comorbidities). Of note, in 2017 one patient from Syria was not listed because his permanent residency was still Syria without the perspective of a long term stay in Germany. Therefor the patient was not listed because of questionable pots-transplant care. Although communication and other factors with potential influence on the compliance and adherence are frequently discussed in our transplant board, we generally believe that adherence and non- adherence are not inalterable conditions and most problems in this respect can be managed with adequate patients support.

A note on the juridic aspect of LTx in migrants could be added. I suppose that by law, all registred migrants have access to Transplant and health care in general. The true difficulty is in patients who very recently arrived and are not resident yet or not registred yet. Were the authors confronted with this situation? Could they highlight this problematic? I remember that a few cases of LTx for ALF (I think for mushroom intoxication) in Germany were reported at a transplant meeting in Belgium (BTS) a few years ago with good outcome. Are the authors aware of these cases? Have they been published/reported/publicized (in literature and/or media)? If yes could it be possible to refer to them?

Answer: In contrast to e.g. Belgium, Germany doesn’t have a ministerial order to transplant unregistered patients in emergency-situations. In the case of unregistered patients presenting in the acute need of a liver transplant, an emergency protocol is sent to the social services department and the patient will receive the care he/she needs. Financial compensation is sorted out during or after the transplant and as a last resort, the social service department will compensate for the treatment. However, we haven’t experienced this situation at our unit yet and only know some cases of acute liver failure in unregistered asylum seekers, where the course of action mentioned above had to be taken { Lehner F. Refugees - new challenge in transplantation. Eurotransplant Annual Meeting 2016}. The manuscript was amended accordingly.

Another difficulty is the access to medications and follow up in case of planned return to a home land without German/European health care. Have the authors been confronted with this scenario and would this play a role in decision-making?

Answer: We have been confronted with this scenario following the transplantation of one tourist who returned to his/her homeland and did not receive adequate post transplant care, although high-level transplant care would have been available and funding was checked and found to be available by our social services team. The negative turn of this case was rather an adherence failure than a systematic problem. In general, patients who receive an organ transplant in Germany and are expected not to receive adequate post-transplant care in the countries or regions they originate from will receive a permanent residency status based on medical necessity.

One might argue that german mig born in Germany might represent a lower risk population compared to recently arrived non german citizens? Can authors comment on that?

Answer: please see the answer to the next question.

Within the mig group, german citizens mig did better than non german. However non german mig did equally well compared to non mig patients. What about german citizens mig versus non mig control?

Answer: German citizens mig did significantly better than non mig patients (p = 0,01) and better than mig patients without german citizenship at time of transplant (p = 0.021). Please see the answer to the next question for our interpretation.

The MV analysis showed no impact of mig status. Did the authors stratify this analysis depending upon the various mig types? German citizens vs non citizens mig etc?

Answer: We thank the reviewer for this comment. We calculated the Cox regression including the mig status and found a significant protective effect of german citizenship a time of transplant. Investigating this interesting finding, we compared the median ages of all three groups and found a significant younger age in the mig 1 group which might explain the significantly better survival in this group. The new analysis was included in the manuscript and discussed accordingly.

Reviewer #2: In this manuscript Bucher et al. investigates the role of recipient history migration on the outcome of liver transplantation in a single center, descriptive cohort study. They did not find differences in outcome between recipient with and without the history of migration.

I read with interest this manuscript and I have few comments and questions to the authors:

The authors perform extensive statistical analysis on this rather small group of patients as 67 patients out of 358 patients had a history of migration. The authors divided them into 10 groups according to the RKI, however they did not give any number of patients assigned to each group. Could you provide this data? Later in their KM analysis they stratified these patients later into two groups.

Answer: The division into 10 groups was in accordance with the official definition of a history of migration through the Robert Koch Institute (RKI) {Robert-Koch-Institut 2008 #1}, Germany’s federal institute for disease-control and prevention. We felt that we needed a well-established and official definition like this as a basis for our analysis. We have included the numbers of assignments to these groups in the results section of the revised manuscript. For the KMC-analysis of the migration status as a factor for the outcome necessitated an aggregation of the groups to reach a sufficient number of patients. Therefor we decided to compare patients who were German citizens at the time of transplant. We feel that the German citizenship best resembled a state of integration in the social system, much better than e.g. the ethnicity, that should have effects on the orientation and functionality of such patients in the healthcare system.

7 patients from mig-group received liver transplantation for acute liver failure. Were in this group also patients who stayed in Germany temporary and moved to their countries after transplantation?

Answer: One patient of this group returned to the home country after transplant.

Did you analyse also graft survival in both groups? If so did you find any differences? It would be also good to know if the compliance after liver transplantation did not differ between two groups.

Answer: Graft survival is defined as the loss of the graft and the need for re-transplantation. We found a low re-transplant rate in both, the mig and the non-mig groups (11 vs. 31 Patients – 16% vs. 11%, non significant). Most of the re-transplants were for acute allograft failure, only 4 % in each group were for chronic allograft failure. Because of these small numbers and since we expect the effects of a history of migration to influence mostly the post discharge phase after LTx, we did not perform a KMC-analysis of graft survival.

From your KM analysis (patients at risks) it seems that majority of mig-group has been transplanted in the later period of your study period. Do you have any explanation for this?

Answer: We have checked the transplant rates during the studied period and do not see any such dynamic (year and transplanted patients with history of migration: 2015:10; 2014:11; 2013:6; 2012:8; 2011:8; 2010:10; 2009:2; 2008:9; 2007:3)

I would be very careful in extrapolating the result of this study on the German situation as the authors did in their conclusions. The retrospective and single-center nature of this study is a serious limitation, so it would be very interesting a have a national data on this topic, before drawing a conclusion that migration background is not a risk factor for the outcome in liver transplantation.

Answer: We also feel that the retrospective and single-center nature of this study is a limitation. However, most of the previous studies on this topic also were single-center retrospective studies. Also, we emphasize in our conclusion, that we mainly present our data because it stands in contrast to the previous literature. We want to express, that the notion, that patients with a history of migration have a worse outcome after LTx might not be true in the German context and that it therefore needs further exploration. Certainly, a nation-wide analysis would be very interesting. Unfortunately, the implementation of a national transplant registry has just begun and data won’t be available until several years in the future. Furthermore, data on a history of migration to our knowledge is not part of the data that is transmitted to this register.

General comment:

As you perform extensive analysis of your relatively small data, the manuscript is not easy to read and its message is somehow lost in the text. I would shorten this manuscript and focus on the most important issues.

Answer: We have reduced data and interpretation of the topic of distance from the transplant center and survival. Manuscript was changed accordingly.

Additional comment:

In my opinion inclusion of Turkey in the Eastern Europe is not correct.

Answer: Certainly the inclusion of Turkey to the Eastern European Region is rather based on the geographic position than on a political or ethnic basis. However, since the social and medical standards (at least of middle and eastern Turkey) are similar to other countries in the Eastern European Region, compared with the other Mediterranean countries that are members of the EU, we felt that for our analysis it was appropriate to include Turkey in the Eastern European Region.

We hope our revised manuscript and these answers are sufficient to improve the quality of our manuscript.

Sincerely, 

Julian Bucher and Markus Guba

---

## [Decision Letter · Decision Letter 1]

7 Oct 2019

Liver transplantation in patients with a history of migration – a German single center comparative analysis

PONE-D-19-18755R1

Dear Dr. Bucher,

We are pleased to inform you that your manuscript has been judged scientifically suitable for publication and will be formally accepted for publication once it complies with all outstanding technical requirements.

With kind regards,

Frank JMF Dor, M.D., Ph.D., FEBS, FRCS

Academic Editor

PLOS ONE

Additional Editor Comments (optional):

Reviewers' comments:

Reviewer's Responses to Questions

**Comments to the Author**

1. If the authors have adequately addressed your comments raised in a previous round of review and you feel that this manuscript is now acceptable for publication, you may indicate that here to bypass the “Comments to the Author” section, enter your conflict of interest statement in the “Confidential to Editor” section, and submit your "Accept" recommendation.

Reviewer #1: All comments have been addressed

Reviewer #2: All comments have been addressed

2. Is the manuscript technically sound, and do the data support the conclusions?

Reviewer #1: Yes

Reviewer #2: Yes

3. Has the statistical analysis been performed appropriately and rigorously? 

Reviewer #1: Yes

Reviewer #2: Yes

4. Have the authors made all data underlying the findings in their manuscript fully available?

Reviewer #1: Yes

Reviewer #2: Yes

5. Is the manuscript presented in an intelligible fashion and written in standard English?

Reviewer #1: Yes

Reviewer #2: Yes

6. Review Comments to the Author

Reviewer #1: (No Response)

Reviewer #2: The authors improved their manuscript substantially according to the reviewers’ comments. I am also satisfied with their answer to my questions. I do not have any further comments.

7. PLOS authors have the option to publish the peer review history of their article (what does this mean?). If published, this will include your full peer review and any attached files.

Reviewer #1: Yes: Jacques Pirenne

Reviewer #2: No

---

## [Editor Report · Acceptance letter]

11 Oct 2019

PONE-D-19-18755R1 

Liver transplantation in patients with a history of migration – a German single center comparative analysis 

Dear Dr. Bucher:

I am pleased to inform you that your manuscript has been deemed suitable for publication in PLOS ONE. Congratulations! Your manuscript is now with our production department. 

With kind regards,

on behalf of

Dr. Frank JMF Dor 

Academic Editor

PLOS ONE